# Evaluation of Human Brucellosis Patients with Post-Treatment Standard Tube Agglutination Test Titers

**DOI:** 10.3390/pathogens14111186

**Published:** 2025-11-19

**Authors:** Aysun Benli, Ayşe Nur Ceylan

**Affiliations:** 1Department of Infectious Diseases and Clinical Microbiology, Istanbul Faculty of Medicine, University of Istanbul, Istanbul 34093, Turkey; 2Department of Medical Microbiology, Basaksehir Çam and Sakura City Hospital, University of Health Science, Istanbul 34480, Turkey; aysenurceylan1011@gmail.com

**Keywords:** *Brucella* spp. brucellosis, standard tube agglutination test, serological follow-up, persistent antibody

## Abstract

**Introduction:** This study was designed to determine the differences between brucellosis patients whose standard tube agglutination test (SAT) titers decreased or not after successful treatment. **Methods:** This retrospective study included patients with a course of antibiotic therapy at least 6 weeks for acute brucellosis or 12 weeks for osteoarticular involvement, and whose post-treatment clinical findings improved. **Results:** The mean age of the 276 patients was 45.2 years, and 50.7% were female. The SAT titer decreased in 166 patients (60%). No significant differences were found in terms of demographical and epidemiological characteristics between the groups. Patients with decreased SAT titers exhibited an elevated pre-treatment erythrocyte sedimentation rate (ESR) and the lymphocytosis was more prevalent. In the non-decreased SAT group, liver enzymes such as aspartate aminotransferase (AST) and alanine aminotransferase (ALT) values after treatment were higher. The initial SAT titer of 1/160 and the pre-treatment rates of anaemia and thrombocytopenia were significantly higher in patients whose SAT titers became negative. Among patients whose SAT titers remained positive, the initial SAT titer was more frequently ≥1/320, and the post-treatment AST value was higher. **Conclusions:** This study showed that a serological response can be obtained with a high ESR and lymphocytosis prior to treatment. It should be noted that SAT negativity cannot be observed immediately in patients with pre-treatment SAT titers ≥ 1/320. The healthcare providers are advised to consider the complete clinical picture without relying solely on serological results.

## 1. Introduction

Brucellosis, the most prevalent zoonosis on a global scale, exhibits a higher incidence in regions where animal husbandry is prevalent. This zoonosis is endemic in the Mediterranean region, Latin America, Africa, the Middle East, and Western Asia. The causative agent of the disease is *Brucella* spp., which is a Gram-negative, oxidase-positive, urease-positive coccobacillus. The bacteria are present in significant quantities in the milk, urine, and birth secretions of infected animals, causing mild or asymptomatic infections. It can be transmitted through direct contact with infected animals’ tissues or secretions, consumption of unpasteurized dairy products, and inhalation of aerosols [1,2]. The symptoms of brucellosis are non-specific. The most common symptoms include fever, chills, fatigue, loss of appetite, weight loss, headache, night sweats, myalgia and arthralgia. In the event of localized involvement, symptoms specific to the affected body part are observed [2,3]. In cases of brucellosis, there is a possibility of mild transaminase elevation and the subsequent development of hepatitis. Additionally, a range of hematological abnormalities, including anemia, lymphocytosis, leukocytosis, leukopenia, and thrombocytopenia, have been documented [4,5]. While the definitive diagnosis of brucellosis necessitates the identification of bacterial growth, the disease is also diagnosed through the combination of positive antibody or antibody titer test results with clinical observations [6,7]. Blood, tissue or bone marrow culture are gold standard methods for the diagnosis of brucellosis, in which the culture resulted negative; but there is a clinical suspicion, and in areas where the culture is not possible, serological tests are crucial for confirming the diagnosis of brucellosis. Agglutination tests are a type of serological test used to assess antibody responses. These tests examine the direct agglutination of bacterial antigens with specific antibodies in the patient’s serum [2,7]. According to the World Health Organization (WHO)’s laboratory criteria for the diagnosis of brucellosis, the isolation of *Brucella* spp. or a standard tube agglutination test (SAT) titer of 1/160 and above is required [8]. Brucellosis is an intracellular pathogen, so it requires long-term combination therapy with multiple antibiotics to prevent recurrence and chronicity. This infectious disease poses significant therapeutic challenges, particularly in endemic areas where the consumption of unpasteurised dairy products is common, resulting in frequent relapses and reinfections.

While it is generally expected that antibodies against *Brucella* spp. will become negative in the blood after successful treatment, it is sometimes observed that antibody levels do not decrease despite clinical and laboratory findings improving in patients [9]. Sometimes it is not always possible to distinguish patients with active disease from those with past infection. Although serological follow-up is not recommended for treatment response, it has been observed in medical practice that repeated treatment courses are applied to patients whose *Brucella* SAT titers do not decrease and remain high. This situation carries the risk of increasing costs, adverse effects that antibiotics can cause, and antibiotic resistance. Other serological tests used in the diagnosis of brucellosis, such as the 2-mercaptoethanol (2-ME) test and the enzyme-linked immunosorbent assay (ELISA) tests, may be less sensitive and specific than the SAT. Since these tests are not used frequently in clinical practice, the role of these tests in serological response is controversial. A comprehensive understanding of these patients, who frequently receive unnecessary treatment, is crucial to averting this predicament. The present study was conceptualised to ascertain whether there exists a discrepancy in terms of epidemiological, clinical, and laboratory characteristics between patients exhibiting decreased *Brucella* SAT titers and those who have non-decreased *Brucella* SAT titers following clinical improvement subsequent to brucellosis treatment.

## 2. Materials and Methods

### 2.1. Data Collection and Patient Inclusion/Exclusion Criteria

The present study was conducted in adult patients diagnosed with acute brucellosis and followed up at the Infectious Diseases outpatient clinic of Mus State Hospital between 2017 and 2020. The province of Mus is an endemic region of brucellosis in the east side of Turkey. The study was planned and data were collected while the authors were at Mus State Hospital for “compulsory medical duty”. The *Brucella* SAT titers of the patients were measured before and after treatment. Demographical, clinical, and laboratory data of the patients was obtained from patient files and the hospital automation system. The following data were collected: age, gender, place of residence (village/district/province), possible risk factors for transmission (unpasteurised dairy product consumption/animal husbandry), presence of osteoarticular involvement, antibiotic treatment regimens, *Brucella* SAT titers, leukocyte, lymphocyte, platelet, hemoglobin, C-reactive protein (CRP), erythrocyte sedimentation rate (ESR), aspartate aminotransferase (AST), and alanine aminotransferase (ALT) values, which were recorded retrospectively. Adult acute brucellosis patients who demonstrated clinical improvement at the end of the treatment, who exhibited compliance with treatment and who had post-treatment SAT titer were included in the study. Patients with chronic brucellosis, reinfection, relapses and also patients with missing information were excluded from the study.

### 2.2. Definitions

Patients with SAT titers ≥ 1/160 and a range of symptoms including fever, chills, fatigue, loss of appetite, weight loss, headache, night sweats, myalgia and arthralgia lasting less than a year were considered to have acute brucellosis. Despite the fact that the majority of these symptoms were not specific to brucellosis, they were considered as brucellosis symptoms because they could not be explained by any other causes. Patients with focal involvement, as demonstrated by bone-joint findings, laboratory findings and radiological findings, were evaluated as patients with osteoarticular involvement [7]. Lymphocytosis was defined as a lymphocyte count above 4000/µL in a complete blood sample, and lymphopenia as a lymphocyte count below 1000/µL. Anaemia was defined as a haemoglobin count below 12 g/dL in female patients and 13 g/dl in male patients. Thrombocytopenia was defined as a platelet count below 150,000/µL. Treatment success was based on clinical improvement and its decision made by the physician. The definition of clinical improvement was regression in initial symptoms/signs and decline in high initial inflammatory markers at the end of the treatment.

### 2.3. Diagnosis of Brucellosis and Follow-Up

The blood culture and species determination for *Brucella* spp. could not be performed because the essential equipment for culture was not available in the region where the study was conducted. The brucellosis was diagnosed by combination of clinical symptoms of brucellosis (mentioned in ‘Definitions’ subheading) and SAT titers ≥ 1/160. The patients were followed up weekly until the end of treatment. During each visit, they were questioned about the recovery status of their initial symptoms, and blood counts and biochemical tests were performed. Post-treatment SAT titers had measured at the visit in which antibiotic therapy was discontinued for all included patients.

### 2.4. Treatment Regimens and Duration of Therapy

The regulation of brucellosis treatment and the follow-up of the patients were carried out by an infectious diseases and clinical microbiology specialist. The patients received standard therapy with at least two drugs (doxycycline + streptomycin or doxycycline + rifampicin) (Doxycycline as Tetradox^®^ from Abdi İbrahim Pharmaceutical Industry and Trade Inc., Istanbul, Turkey; Streptomycin as Streptomisin sülfat^®^, from I.E. Ulagay Pharmaceutical Industry Turk Inc., Istanbul, Turkey) as recommended by some experts and WHO [3,4]. Clinician-based treatment was administered to patients with osteoarticular involvement and to some patients predicted to be unresponsive to therapy by adding a third agent to doxycycline + rifampicin. The doses and the administration route of the drugs used in the treatment of brucellosis are as follows: doxycycline 2 × 100 mg orally, rifampin 1 × 600 mg orally, streptomycin 1 × 1 gm intramuscularly, ciprofloxacin 2 × 500 mg orally, and trimethoprim-sulfamethoxazole 2 × 160–800 mg orally. Patients were treated at least 6 weeks for acute brucellosis and 12 weeks for brucellosis with osteoarticular involvement until their initial symptoms resolved. In cases where streptomycin was part of the combination treatment, it was administered for a duration of three weeks, with the other drugs being completed for the designated period. Treatment regimens were determined according to the patients’ characteristics, additional diseases, and the presence of drug allergies. The doses of antibiotics adjusted according to the glomerular filtration rate of the patients if necessary.

### 2.5. Standard Tube Agglutination Test

The agglutination titers for *Brucella* spp. in patients’ serum samples were determined using the SAT at the Mus State Hospital Central Laboratory. The test was performed using serum dilutions ranging from 1/80 to 1/2560, and titers of 1/160 or higher were considered positive. These tests were performed by a medical microbiology specialist. Agglutination titers were evaluated at a range from 1/80 to 1/2560.

### 2.6. Statistical Analysis

In those who demonstrated clinical improvement following an adequate duration of brucellosis treatment, patients whose *Brucella* SAT titer remained constant or increased were categorised as those whose *Brucella* SAT did not decrease; and patients whose *Brucella* SAT titer decreased or whose *Brucella* SAT titer was determined as negative were categorised as those whose *Brucella* SAT titer decreased. The characteristics of the patients divided into two groups were evaluated using statistical tests. The analysis of continuous variables was conducted using independent samples *t*-test if the distribution was normal, and Mann–Whitney U test if the distribution was not normal. For the analysis of categorical variables, the χ^2^ test or Fisher’s exact test was employed. Furthermore, patients whose *Brucella* SAT became completely negative and those whose *Brucella* SAT did not become negative were also evaluated within themselves with the same statistical tests, as detailed above. Statistically significant results were defined as those with a *p* value less than 0.05. Descriptive statistics were given as numbers and percentages for categorical variables, and mean and standard deviation for numerical variables. The statistical analysis of the study data was conducted using IBM SPSS Statistics for Windows, Version 29.0 (Statistical Package for the Social Sciences, IBM Corp., Armonk, NY, USA).

### 2.7. Ethical Approval

This study was performed in line with the principles of the Declaration of Helsinki. Ethical approval was obtained from the ethics committee of Basaksehir Cam and Sakura City Hospital (Protocol code: 2022/394, Date: 14 December 2022).

## 3. Results

A total of 276 patients diagnosed with acute brucellosis were included in the study. The mean age of the patients was 45.2 years, and 50.7% of them were female. When the possible risk factors for the transmission of the disease were examined, most of the patients were found to be deal with animal husbandry (54.7%), 21% consumed unpasteurised milk products, and the transmission route was unknown for 24.3%. This information was obtained by questioning the patients; detailed investigations could not be conducted to determine the real source of the infection. However, in the region where the study was conducted, patients and their families raised sheep and cows intensively, assisted with the animals’ deliveries and reported frequent miscarriages due to brucellosis. They also frequently consumed unpasteurised dairy products obtained from these animals. The majority of patients resided in rural areas (52.2%), and osteoarticular involvement was observed in 31 (11.2%) patients, with 20 (7.2%) cases of spondylodiscitis. The most prevalent antibiotic combination was doxycycline + rifampicin (50.4%), followed by doxycycline + streptomycin (22.8%) (Table 1 and Table 2). The laboratory values of the patients before and after treatment are shown also (Table 1 and Table 2). When the data from all patients was considered, the mean values of leukocyte, ESR, CRP, AST, and ALT, which were higher before treatment, decreased after brucellosis treatment. Half of the patients had a pre-treatment *Brucella* SAT titer of 1/640 or higher. Furthermore, lymphocytosis, lymphopenia, anaemia and thrombocytopenia were observed in 6.2%, 5.1%, 14.5% and 3.3% of patients, respectively.

A decline in *Brucella* SAT titers was observed in 166 (60%) patients following brucellosis treatment, while no such decline was noted in the remaining 110 (40%) patients. Upon evaluation of the two groups, it was observed that they were comparable in terms of age, gender distribution, potential risk factors for transmission, place of residence, presence of osteoarticular involvement, and antibiotic treatment regimens applied. No statistically significant differences were identified for all these (Table 1). However, pre-treatment ESR was found to be elevated in patients with decreased *Brucella* SAT titers (*p* = 0.048). Furthermore, lymphocytosis was found to be more prevalent in the decreased SAT group prior to treatment (*p* = 0.045) (Table 1). Among patients without a decrease in SAT titers, mean post-treatment AST and ALT levels were significantly higher compared with those in the decreased SAT group (*p* < 0.001 and *p* = 0.049, respectively). Although not statistically significant, a similar tendency was observed before treatment, with patients in the non-decreased SAT group showing higher baseline AST and ALT levels than those with decreased SAT titers (Table 1).

In the evaluation made between the patients whose SAT titers became negative (n = 81) and all the remaining patients (n = 195), the rate of pre-treatment SAT titer being 1/160 and the rates of anaemia and thrombocytopenia of the patients before the treatment were significantly higher in those whose SAT titers became negative (*p* < 0.001, *p* = 0.019 and *p* = 0.021, respectively) (Table 2). In patients whose SAT titers remained positive, initial SAT titers were more frequently ≥1/320, and post-treatment AST values were higher (*p* < 0.001 and *p* = 0.039, respectively). Although not statistically significant, osteoarticular involvement was higher in patients whose SAT titers did not become negative (Table 2).

## 4. Discussion

In this study, *Brucella* SAT titers remained positive in a substantial proportion of patients after treatment. Higher pre-treatment SAT titers (≥1/320) were associated with persistent SAT positivity, while patients with lower pre-treatment SAT titers (≤1/160) were more likely to achieve serological negativity. AST and ALT levels tended to remain elevated in patients whose SAT titers did not decrease.

The research following the *Brucella* SAT titer after treatment has demonstrated that the SAT remains positive for two years or even longer [9,10]. In a study, a decrease in the *Brucella* SAT titer below 1/320 after treatment was accepted as serological cure; young age, female gender, doxycycline-containing treatment, and use of three or more antibiotics in treatment were shown to be associated with serological cure [9]. The implication of this article is the same as ours, patient monitoring with SAT results in overdiagnosis and unnecessary exposure of patients to anti-*Brucella* treatment. In a separate study conducted in Turkey, researchers discussed the role of SAT in monitoring patients. The SAT negativity following brucellosis treatment was observed in 18% of patients, lower than our study. Furthermore, female gender, the absence of fever, the absence of arthralgia, and pre-treatment SAT titers below 1/160 were found to be associated with serological cure [11]. However, as indicated in the study by Copur et al., in our study, SAT of 1/160 was found to be associated with serological response. It was observed that demographical characteristics of the patients, osteoarticular involvement and treatment regimens did not significantly affect the *Brucella* SAT negativity in our study. However, although not statistically significant, it was observed in our study that the age of patients whose SAT titers did not decrease was higher and osteoarticular involvement was higher in patients whose SAT titers did not become negative. As was evidenced in our study, another study has also demonstrated that *Brucella* serology does not correlate with clinical outcomes after treatment [12]. In that study, no significant correlations were found between antibody titers of *Brucella* spp. and clinical cure, culture positivity and other laboratory results [12]. In a study in China, researchers followed patients up with acute brucellosis for six months after treatment. Furthermore, 32% of patients exhibited positive SAT titers, while 19% continued to report arthralgia. That study emphasised that the positivity of the SAT after treatment and alterations in antibody titers can be utilised for the purpose of distinguishing patients with chronic brucellosis [13].

In a large series of patients, 51.3% of the patients were male, and the median age was 41 in men and 45 in women, which was similar to our study. In addition, 32% of the patients in this study had at least one hematological abnormality. As in our study, anaemia was the most common hematological abnormality in this study [14]. It is interesting that SAT negativity is more common in patients with hematological abnormalities in our study.

A series of studies have been conducted on the serological follow-up of brucellosis, with a view to determine which test may be more sensitive. In a study comparing the 2-ME test with SAT, SAT positivity at six months after treatment was found to be 23.4%, while 2-ME test positivity was found to be 29.1%. The serological cure for both tests with lower titers were higher than those with higher titers, as shown in our study. For the SAT, serological cure (a decrease to 1/160 and below) was higher at 1/640 and below [15]. In another study, the 2-ME test was found to be superior to SAT in determining the duration of appropriate antibiotic treatment, and its negative result was recommended as a test to exclude chronic brucellosis [10]. And the researchers of that study found that, following a period of one and a half years, the rate of 2-ME agglutination test positivity was 4% while SAT positivity was 48% [9]. In the serological follow-up of brucellosis, IgG developed against the cytoplasmic proteins of *Brucella* spp. and, as determined by ELISA, was found to be superior to SAT and 2-ME tests [16]. A study investigating the role of *Brucella* enzyme immunoassay (EIA) and 2-ME tests in brucellosis follow-up revealed that IgM is a safer diagnostic and treatment follow-up option for acute brucellosis, with the potential to prevent approximately half of unnecessary treatments [17]. A study showing a significant difference in pre- and post-treatment IgM titers of patients diagnosed with brucellosis revealed that no such difference was observed in IgG levels, particularly among chronic patients. Accordingly, the study emphasised that combining IgG and IgM tests for diagnosing and monitoring brucellosis would produce more accurate results [18]. Following the administration of effective treatment for brucellosis, the presence of IgM antibodies may persist at low titers for a period of months or years. The presence of *Brucella* antibodies over an extended period can complicate the distinction between recurrence and prior infection. Consequently, a study indicated that high IgG avidity may be beneficial in excluding active infection [19]. The sustained antibody response following brucellosis treatment may be attributable to the prolonged persistence of long-lived plasma cells, which continue to produce antibodies even after a period of months or years in the absence of antigen [20]. Additionally, memory B cells may be reactivated by cross-stimulation with other bacteria or low levels of *Brucella* antigen residue, leading to the continued production of antibody [21].

*Brucella* PCR is a useful monitoring tool for the evaluation of treatment efficacy and is a more reliable indicator of the patients’ health status, helping clinicians make more informed and better diagnoses [22]. A recent article has determined new biological markers for the diagnosis and follow-up of brucellosis. The study demonstrated that the decrease in ABI3, CLEC12B, PPP2R4 protein levels and the increase in DDIT4L, PIAS4, IDO protein levels, as detected by a microarray study, can be used to evaluate the treatment response [23].

In the diagnosis of brucellosis, SAT is a test that measures both IgG and IgM antibodies, and is therefore regarded as the primary diagnostic test. However, as it can remain at high titers for an extended period after treatment, it may not be sufficient on its own to determine whether the condition is acute or chronic, to evaluate treatment effectiveness, or to detect relapses and focal infections. In regions where brucellosis is endemic, the presence of persistent specific IgG antibodies that may already exist in the serum of the population may lead to confusion in serological diagnosis. Methods that allow the measurement of IgM values or new biomarkers are therefore needed. In the future, the advent of new biomarkers and their widespread use are poised to usher in a new era in the diagnosis and follow-up of brucellosis.

The study is subject to the following limitations. Firstly, the study only includes patients from one hospital in an endemic region, which limits generalizability to populations in different geographic and clinical settings. Secondly, since other serological tests other than SAT were not studied, we cannot talk about the functionality of other tests in this patient population. Thirdly, due to the absence of prolonged patient monitoring, the availability of data concerning the long-term alterations in SAT titers remains limited. And lastly, blood culture and species determination for *Brucella* spp. could not be performed. Multicenter and larger studies are needed to identify potential risk factors related to serological unresponsiveness.

## 5. Conclusions

In conclusion, our study has demonstrated that in some patients, despite having similar demographical, clinical and treatment characteristics, brucellosis SAT titers may continue to remain positive after treatment. It should be kept in mind that in patients with ongoing SAT positivity, AST and ALT values may remain higher after treatment, and that SAT negativity may not be observed immediately after treatment in those with pre-treatment SAT titers ≥ 1/320. Apart from this, no demographical or clinical differences were detected between patients whose SAT became negative or remained positive. SAT was examined after treatment for this study and routine post-treatment SAT monitoring is unnecessary unless clinically indicated. The improvement of clinical findings in patients given brucellosis treatment for appropriate periods and doses will be sufficient to terminate the treatment. According to the results of this study, it is not recommended to re-initiate treatment in patients who have a high SAT titer at any time but do not have clinical signs and symptoms of brucellosis. As a result, the serological follow-up should be interpreted in conjunction with clinical and biochemical improvement and ideally supported by bacteriological confirmation when possible. And comprehensive studies are needed to use other microbiological methods in follow-up brucellosis and not rely solely on SAT.

## Figures and Tables

**Table 1 pathogens-14-01186-t001:** Demographical, epidemiological, laboratory and treatment characteristics of patients with and without decreased SAT titer after *brucellosis* treatment.

	All Patients (n = 276)	Patients with Decreased SAT (n = 166)	Patients Without Decreased SAT (n = 110)	*p* Value
**Age** (mean ± SD)	45.2 ± 14.5	44.2 ± 15	46.7 ± 13.7	0.125
**Gender** (n, %)				0.659
Female	140 (50.7%)	86 (51.9%)	54 (49.1%)	
Male	136 (49.3%)	80 (48.1%)	56 (50.9%)	
**Possible risk factors for transmission** (n, %)				0.492
Consumption of unpasteurised dairy products	58 (21%)	37 (22.2%)	21 (19%)	
Animal husbandry	151 (54.7%)	86 (51.8%)	65 (59%)	
Unknown	67 (24.3%)	43 (25.9%)	24 (21.8%)	
**Place of residence** (n, %)				0.384
Village	144 (52.2%)	81 (48.7%)	63 (57.2%)	
District	40 (14.5%)	26 (15.6%)	14 (12.7%)	
Province	92 (33.3%)	59 (35.5%)	33 (30%)	
**Osteoarticular involvement** (n, %)	31 (11.2%)	19 (11.4%)	12 (10.9%)	0.890
Spondylodiscitis	20 (7.2%)	14 (8.4%)	6 (5.4%)	0.350
Arthritis	11 (3.9%)	5 (3%)	6 (5.4%)	0.442
**Treatment regimens** (n, %)				0.738
Doxycycline + rifampicin	140 (50.7%)	87 (52.4%)	53 (48.1%)	
Doxycycline + streptomycin	63 (22.8%)	35 (21%)	28 (25.4%)	
Doxycycline + rifampicin + streptomycin	53 (19.2%)	33 (19.8%)	20 (18.1%)	
Doxycycline + rifampicin + ciprofloxacin	2 (0.7%)	1 (0.6%)	1 (0.9%)	
Doxycycline + rifampicin + trimethoprim-sulfamethoxazole	18 (6.5%)	10 (6%)	8 (7.2%)	
**Pre-treatment laboratory results**				
Leukocyte, cell/µL (mean ± SD)	7493 ± 2206	7471 ± 2155	7527 ± 2291	0.788
ESR (mean ± SD)	29.2 ± 22.5	31.5 ± 23	25.8 ± 21.5	**0.048**
CRP, mg/L (mean ± SD)	23.2 ± 36.3	24.6 ± 39.4	21.3 ± 31.4	0.577
AST, U/L (mean ± SD)	28 ± 32	27.8 ± 37.9	28.3 ± 21.4	0.232
ALT, U/L (mean ± SD)	30 ± 28	29.2 ± 27.7	32 ± 29.7	0.213
Lymphocytosis (n, %)	17 (6.2%)	14 (8.4%)	3 (2.7%)	**0.045**
Lymphopenia (n, %)	14 (5.1%)	8 (4.8%)	6 (5.4%)	0.814
Anemia (n, %)	40 (14.5%)	26 (15.6%)	14 (12.7%)	0.498
Thrombocytopenia (n, %)	9 (3.3%)	6 (3.6%)	3 (2.7%)	1
SAT = 1/160 (n, %)	73 (26.4%)	38 (22.9%)	35 (31.8%)	0.100
SAT ≥ 1/320 (n, %)	203 (73.5%)	128 (77.1%)	75 (68.1%)	0.100
SAT ≥ 1/640 (n, %)	138 (50%)	85 (51.2%)	53 (48.1%)	0.623
SAT ≥ 1/1280 (n, %)	90 (32.6%)	54 (32.5%)	36 (32.7%)	0.973
SAT ≥ 1/2560 (n, %)	4 (1.4%)	1 (0.6%)	3 (2.7%)	0.305
**Post-treatment laboratory results**				
Leukocyte, cell/µL (mean ± SD)	6734 ± 1839	6987 ± 1826	6435 ± 1818	0.062
ESR (mean ± SD)	19.4 ± 17.1	19.2 ± 18	19.5 ± 16.3	0.607
CRP, mg/L (mean ± SD)	6.8 ± 14.4	7.1 ± 13.8	6.5 ± 15.2	0.756
AST, U/L (mean ± SD)	19.9 ± 7.6	18.2 ± 5.1	22 ± 9.4	**<0.001**
ALT, U/L (mean ± SD)	20.3 ± 13.9	18.2 ± 7.5	22.9 ± 18.5	**0.049**

SAT: standard tube agglutination test, SD: standard deviation, ESR: erythrocyte sedimentation rate, CRP: C-reactive protein, AST: Aspartate aminotransferase, ALT: Alanine aminotransferase.

**Table 2 pathogens-14-01186-t002:** Demographical, epidemiological, laboratory and treatment characteristics of patients with negative or positive SAT titer after brucellosis treatment.

	All Patients (n = 276)	Patients with Negative SAT (n = 81)	Patients with Positive SAT (n = 195)	*p* Value
**Age** (mean ± SD)	45.2 ± 14.5	45.4 ± 15.8	45.1 ± 14	0.894
**Gender** (n, %)				0.301
Female	140 (50.7%)	36 (44.4%)	100 (51.2%)	
Male	136 (49.3%)	45 (55.5%)	95 (48.7%)	
**Possible risk factors for transmission** (n, %)				0.255
Consumption of unpasteurised dairy products	58 (21%)	16 (19.7%)	42 (21.5%)	
Animal husbandry	151 (54.7%)	40 (49.3%)	111 (56.9%)	
Unknown	67 (24.3%)	25 (30.8%)	42 (21.5%)	
**Place of residence** (n, %)				0.365
Village	144 (52.2%)	37 (45.6%)	107 (54.8%)	
District	40 (14.5%)	14 (17.2%)	26 (13.3%)	
Province	92 (33.3%)	30 (37%)	62 (31.7%)	
**Osteoarticular involvement** (n, %)	31 (11.2%)	5 (6.1%)	26 (13.3%)	0.086
Spondylodiscitis	20 (7.2%)	4 (4.9%)	16 (8.2%)	0.340
Arthritis	11 (3.9%)	1 (1.2%)	10 (5.1%)	0.678
**Treatment regimens** (n, %)				0.835
Doxycycline + rifampicin	140 (50.7%)	43 (53%)	97 (49.7%)	
Doxycycline + streptomycin	63 (22.8%)	16 (19.7%)	47 (24.1%)	
Doxycycline + rifampicin + streptomycin	53 (19.2%)	17 (20.9%)	36 (18.4%)	
Doxycycline + rifampicin + ciprofloxacin	2 (0.7%)	0 (0)	2 (1%)	
Doxycycline + rifampicin + trimethoprim-sulfamethoxazole	18 (6.5%)	5 (6.1%)	13 (6.6%)	
**Pre-treatment laboratory results**				
Leukocyte, cell/µL (mean ± SD)	7493 ± 2206	7210 ± 2126	7612 ± 2234	0.085
ESR (mean ± SD)	29.2 ± 22.5	32.1 ± 24.2	28.2 ± 21.8	0.282
CRP, mg/L (mean ± SD)	23.2 ± 36.3	25.7 ± 49.1	22.2 ± 29.9	0.708
AST, U/L (mean ± SD)	28 ± 32	18.2 ± 5.2	20.6 ± 8.2	0.311
ALT, U/L (mean ± SD)	30 ± 28	30 ± 32.4	30.5 ± 26.8	0.353
Lymphocytosis (n, %)	17 (6.2%)	4 (4.9%)	13 (6.6%)	0.785
Lymphopenia (n, %)	14 (5.1%)	6 (7.4%)	8 (4.1%)	0.365
Anemia (n, %)	40 (14.5%)	18 (22.2%)	22 (11.2%)	**0.019**
Thrombocytopenia (n, %)	9 (3.3%)	6 (7.4%)	3 (1.5%)	**0.021**
SAT = 1/160 (n, %)	73 (26.4%)	38 (46.9%)	35 (17.9%)	**<0.001**
SAT ≥ 1/320 (n, %)	203 (73.5%)	43 (53%)	160 (82%)	**<0.001**
SAT ≥ 1/640 (n, %)	138 (50%)	19 (23.4%)	119 (61%)	**<0.001**
SAT ≥ 1/1280 (n, %)	90 (32.6%)	11 (13.5%)	79 (40.5%)	**<0.001**
SAT ≥ 1/2560 (n, %)	4 (1.4%)	0 (0)	4 (2%)	0.324
**Post-treatment laboratory results**				
Leukocyte, cell/µL (mean ± SD)	6734 ± 1839	6687 ± 1421	6752 ± 1976	0.869
ESR (mean ± SD)	19.4 ± 17.1	21.2 ± 18.3	18.8 ± 16.8	0.581
CRP, mg/L (mean ± SD)	6.8 ± 14.4	6.5 ± 7.4	6.9 ± 16.2	0.518
AST, U/L (mean ± SD)	19.9 ± 7.6	18.2 ± 5.2	20.6 ± 8.2	**0.039**
ALT, U/L (mean ± SD)	20.3 ± 13.9	18 ± 7.6	21.2 ± 15.4	0.165

SAT: standard tube agglutination test, SD: standard deviation, ESR: erythrocyte sedimentation rate, CRP: C-reactive protein, AST: Aspartate aminotransferase, ALT: Alanine aminotransferase.

## Data Availability

The original contributions presented in this study are included in the article. Further inquiries can be directed to the corresponding author.

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
