# Peer review of "Evaluation of Human Brucellosis Patients with Post-Treatment Standard Tube Agglutination Test Titers"

_pathogens, 2025, doi:10.3390/pathogens14111186_

Round 1
Reviewer 1 Report
Comments and Suggestions for Authors
The study addresses an important and clinically relevant question regarding the interpretation of post-treatment SAT titers in brucellosis. Persistent seropositivity after clinical recovery often leads to unnecessary antibiotic courses, and this work contributes valuable insight into the relationship between serological dynamics and biochemical or hematological parameters. The manuscript is clear, methodologically sound, and well-structured, but certain aspects could be strengthened to improve scientific rigor and readability. The study is valuable and well-conceived, but improvements in methodological transparency, statistical reporting, and discussion depth are necessary before publication.
Major Comments
1-The retrospective design is appropriate, but details about patient selection criteria, exclusion parameters (e.g., relapse, reinfection, comorbidities), and diagnostic confirmation methods (culture, clinical criteria) should be more explicitly described to ensure reproducibility and minimize selection bias. It would be beneficial to clarify how “clinical improvement” was defined, whether standardized criteria were used, or if it was based on physician judgment.
2-The statistical tests applied to compare groups are not clearly stated. The manuscript should specify the tests used (e.g., t-test, Mann–Whitney U, χ²) and include p-values for all significant findings. A table summarizing demographic, clinical, and laboratory comparisons between groups (decreased vs. non-decreased SAT titers) would help the reader visualize the data more effectively.
3-The authors correctly highlight that SAT titers may remain high even after adequate therapy. However, the biological rationale (e.g., persistence of IgG antibodies, immunological memory) should be discussed to contextualize the findings.
4-The discussion could be strengthened by comparing results with other published studies that assessed post-treatment serological profiles in brucellosis.
5-The conclusion should emphasize that serological follow-up must be interpreted in conjunction with clinical and biochemical improvement, and ideally supported by bacteriological confirmation when possible.
6-The statement “SAT should not be measured in clinical practice” might be too absolute; it could be rephrased to suggest that routine post-treatment SAT monitoring is unnecessary unless clinically indicated.
Minor Comments
1-The abstract is generally clear but could be condensed by removing redundant phrasing.
2-The manuscript would benefit from careful language editing to correct minor grammatical inconsistencies and revise the abbreviations used throughout (e.g., SAT, ESR, AST, ALT).
3-Some references to national testing practices (“in our country”) should be replaced by neutral phrasing for an international audience.
Author Response
Response to Reviewer 1 Comments
Thank you very much for taking the time to review this manuscript. Please find the detailed responses below and the corresponding revisions/corrections highlighted green in the re-submitted files. The corrections proposed by other reviewers are highlighted by different colours. Thank you again for your suggestions on how to improve the readability and value of the article, and for giving us the opportunity to implement them. All your suggestions were reviewed point-by-point, and the manuscript was revised taking them into consideration. The method section of the article was improved, the statistical reporting section was detailed, the tables were rearranged, and finally, discussions supporting the results were added to the article. Additionally, the text was improved in general and spelling errors were corrected. If you have any suggestions when you review the manuscript, please do not hesitate to let us know, we would be grateful if you could share them with us.
Point-by-point response to Comments and Suggestions for Authors
Comments 1: [The retrospective design is appropriate, but details about patient selection criteria, exclusion parameters (e.g., relapse, reinfection, comorbidities), and diagnostic confirmation methods (culture, clinical criteria) should be more explicitly described to ensure reproducibility and minimize selection bias. It would be beneficial to clarify how “clinical improvement” was defined, whether standardized criteria were used, or if it was based on physician judgment.]
Response 1: Thank you for pointing this out. We agree with this comments. The patient inclusion/exclusion criteria was added to ‘Data collection’ subheading part in methods section. It was added as ‘Adult acute brucellosis patients who demonstrated clinical improvement at the end of the treatment, who exhibited compliance with treatment and who had post-treatment SAT titer were included in the study. Patients with chronic brucellosis, reinfection, relapses and also patients with missing information were excluded from the study.’ in page 3, lines 95-99. The diagnostic confirmation method explained in seperate subheading as the other reviewer requested too, as ‘Diagnosis of brucellosis and follow-up’ in page 3, lines 115-123. Clinical improvement definition was added to methods section of the article under the ‘Definitions’ subheading, in page 3, lines 111-114 as ‘Treatment success was based on clinical improvement and its decision made by the physician. The definition of clinical improvement was regression in initial symptoms/signs and decline in high initial inflammatory markers at the end of the treatment.’
Comments 2: [The statistical tests applied to compare groups are not clearly stated. The manuscript should specify the tests used (e.g., t-test, Mann–Whitney U, χ²) and include p-values for all significant findings. A table summarizing demographic, clinical, and laboratory comparisons between groups (decreased vs. non-decreased SAT titers) would help the reader visualize the data more effectively.]
Response 2: Thank you for pointing this out. We agree with this comments. The statistical tests applied to compare the groups were clearly stated in the manuscript in page 4, lines 152-159 as ‘The characteristics of the patients divided into two groups were evaluated using statistical tests. The analysis of continuous variables was conducted using independent samples t-test if the distribution was normal, and Mann-Whitney U test if the distribution was not normal. For the analysis of categorical variables, the χ2 test or Fisher's exact test was employed. Furthermore, patients whose Brucella SAT became completely negative and those whose Brucella SAT did not become negative were also evaluated within themselves with the same statistical tests, above. Statistically significant results were defined as those with a p value less than 0.05.’ The p values for all significant findings were added to manuscript as you suggested and highlighted in the text in page 5, line 195, 196, 205, 206, 207 and 208. The tables were divided to help the reader visualize the data more effectively. All demographical, epidemiological, laboratory and treatment characteristics of the patients gathered into one table dividing according to decreased/non-decreased SAT or negative/positive SAT. You can see the changes in pages 5-7, lines 210-219.
Comments 3: [The authors correctly highlight that SAT titers may remain high even after adequate therapy. However, the biological rationale (e.g., persistence of IgG antibodies, immunological memory) should be discussed to contextualize the findings.]
Response 3: Thank you for pointing this out. We agree with this comment. As you suggested, we have included the probable pathobiological reasons why SAT titres remain high even after adequate treatment in the discussion section, in page 8, lines 281-286 as ‘The sustained antibody response following brucellosis treatment may be attributable to the prolonged persistence of long-lived plasma cells, which continue to produce antibodies even after a period of months or years in the absence of antigen20. Additionally, memory B cells may be reactivated by cross-stimulation with other bacteria or low levels of Brucella antigen residue, leading to the continued production of antibody21.’
Comments 4: [The discussion could be strengthened by comparing results with other published studies that assessed post-treatment serological profiles in brucellosis.]
Response 4: Thank you for pointing this out. We agree with this comment. The discussions supporting the results were added to the manuscript. The discussion was strengthened by comparing results with other published studies that assessed post-treatment serological profiles in brucellosis. You can see the additional discussion parts in page 7, lines 232-234 as ‘In a separate study conducted in Turkey, researchers discussed the role of SAT in monitoring patients. The SAT negativity following brucellosis treatment was observed in 18% of patients, lower than our study.’; in pages 7-8, lines 243-251 as ‘As was evidenced in our study, another study has also demonstrated that Brucella serology does not correlate with clinical outcomes after treatment12. In that study, no significant correlations were found between antibody titers of Brucella spp. and clinical cure, culture positivity and other laboratory results12. In a study in China, researchers followed up patients with acute brucellosis for six months after treatment. Furthermore, 32% of patients exhibited positive SAT titers, while 19% continued to report arthralgia. That study emphasised that the positivity of the SAT after treatment and alterations in antibody titers can be utilised for the purpose of distinguishing patients with chronic brucellosis13.’
Comments 5: [The conclusion should emphasize that serological follow-up must be interpreted in conjunction with clinical and biochemical improvement, and ideally supported by bacteriological confirmation when possible.]
Response 5: Thank you for pointing this out. We agree with this comment. We have accordingly revised to emphasize this point. The expression you suggested has been added to conclusion section as ‘As a result, the serological follow-up should be interpreted in conjuction with clinical and biochemical improvement and ideally supported by bacteriological confirmation when possible.’, in page 9, lines 325-327.
Comments 6: [The statement “SAT should not be measured in clinical practice” might be too absolute; it could be rephrased to suggest that routine post-treatment SAT monitoring is unnecessary unless clinically indicated.]
Response 6: Thank you for pointing this out. We agree with this comment. We have accordingly revised to emphasize this point. The expression you suggested has been changed as ‘routine post-treatment SAT monitoring is unnecessary unless clinically indicated’ instead of ‘should not be measured in clinical practice’ in page 9, lines 320-321.
Comments 7: [The abstract is generally clear but could be condensed by removing redundant phrasing.]
Response 7: Thank you for pointing this out. We agree with this comment. Therefore, the abstract has been revised. Unnecessary repetitions and redundant phrasing have been removed and some additions have been highlighted. These changes can be found in page 1, lines 11-29.
Comments 8: [The manuscript would benefit from careful language editing to correct minor grammatical inconsistencies and revise the abbreviations used throughout (e.g., SAT, ESR, AST, ALT).]
Response 8: Thank you for pointing this out. We agree with this comments. Minor grammatical errors have been corrected and abbreviations revised in the manuscript. For example, in page 4, line 184. Due to different coloured highlightings, other corrections may not be visible.
Comments 9: [Some references to national testing practices (“in our country”) should be replaced by neutral phrasing for an international audience.]
Response 9: Thank you for pointing this out. We agree with this comment. Therefore, the phrase 'in our country' was removed from three places in the article. One of them was changed as ‘clinical practice’ in page 2, lines 73-74. And the other was changed as ‘Turkey’ in page 7, line 233.
Reviewer 2 Report
Comments and Suggestions for Authors
The study is descriptive and retrospective and analyzed differences between patients who did or did not show a reduction in SAT (standard agglutination test) values after treatment for brucellosis.
-Some information needs to be added to the manuscript:
How was the diagnosis of brucellosis made? Has the causative agent been identified?
How long were the patients monitored? What criteria were used to perform the analyses after treatment? Immediately after the end of the therapeutic regimen?
-In addition, some changes should be made to improve understanding of the results:
The study did not evaluate the potential risk factor for brucellosis transmission. What was presented was the possible source of infection (consumption of dairy products, animal husbandry). Please amend
I suggest replacing “Microbiological method” (line 104) with “Standard Agglutination Tube Test,” as only this test is being described.
The tables need to be improved. I suggest increasing the width of the tables so that each piece of information is on a single line, making it easier to view the results.
In the discussion, authors should make it clear when they are discussing the results of the present study. For example, in lines 221-224, are the results from other studies?
Check for minor typographical errors: datas (line 70), gr (line 93).
Check that Brucella should be in italics when applicable.
Author Response
Response to Reviewer 2 Comments
Thank you very much for taking the time to review this manuscript. Please find the detailed responses below and the corresponding revisions/corrections highlighted blue in the re-submitted files. The corrections proposed by other reviewers are highlighted by different colours. Thank you again for your suggestions on how to improve the readability and value of the article, and for giving us the opportunity to implement them. All your suggestions were reviewed point-by-point, and the manuscript was revised taking them into consideration. The methods, results section of the article and the tables were improved with your comments. Additionally, the text was improved in general and spelling errors were corrected. If you have any suggestions when you review the manuscript, please do not hesitate to let us know, we would be grateful if you could share them with us.
Point-by-point response to Comments and Suggestions for Authors
Comments 1: [How was the diagnosis of brucellosis made? Has the causative agent been identified?]
Response 1: Thank you for pointing this out. We have accordingly revised to emphasize this point.
Additional information about this suggestion explained in method section of the article as a seperate subheading ‘Diagnosis of brucellosis and follow-up’ in page 3, lines 116-119. ‘The blood culture and species determination for Brucella spp. could not be performed because the essential equipman for culture was not available in the region where the study has conducted. The brucellosis diagnosed by combination of clinical symptoms of brucellosis (mentioned in ‘Definitions’ subheading) and SAT titers ≥1/160.’
Comments 2: [How long were the patients monitored? What criteria were used to perform the analyses after
treatment? Immediately after the end of the therapeutic regimen?]
Response 2: Thank you for pointing this out. We have accordingly revised to emphasize this point.
Additional information about this suggestion explained in method section of the article as a seperate subheading ‘Diagnosis of brucellosis and follow-up’ in page 3, lines 119-123. ‘The patients were followed up weekly until the end of treatment. During each visit, they were questioned about the recovery status of their initial symptoms, and blood counts and biochemical tests were performed. Post-treatment SAT titers had measured at the visit in which antibiotic therapy was discontinued for all included patients.’
Comments 3: [The study did not evaluate the potential risk factor for brucellosis transmission. What was presented was the possible source of infection (consumption of dairy products, animal husbandry). Please amend.]
Response 3: Thank you for pointing this out. We have accordingly revised to emphasize this point. We emphasized the the potential risk factor for brucellosis transmission in page 4, lines 169-178 as ‘A total of 276 patients diagnosed with acute brucellosis were included in the study. The mean age of the patients was 45.2 years, and 50.7% of them were female. When the possible risk factors for the transmission of the disease were examined, most of the patients were found to be deal with animal husbandry (54.7%), 21% consumed unpasteurized milk products, and the transmission route was unknown for 24.3%. This information was obtained by questioning the patients, detailed investigations could not be conducted to determine the real source of the infection. However, in the region where the study was conducted, patients and their families raised sheep and cows intensively, assisted with the animals' deliveries and reported frequent miscarriages due to brucellosis. They also frequently consumed unpasteurised dairy products obtained from these animals.’
Comments 4: [I suggest replacing “Microbiological method” (line 104) with “Standard Agglutination Tube Test,” as only this test is being described.]
Response 4: Thank you for pointing this out. We agree with this comment. We removed the ‘Microbiological method’ and replaced it with ‘Standard Agglutination Tube Test’ in page 4, line 141.
Comments 5: [The tables need to be improved. I suggest increasing the width of the tables so that each piece of information is on a single line, making it easier to view the results.]
Response 5: Thank you for pointing this out. We agree with this comment. The tables were improved according to your suggestion and the other reviewer, too. They were divided to help the reader visualize the data more effectively. All demographical, epidemiological, laboratory and treatment characteristics of the patients gathered into one table dividing according to decreased/non-decreased SAT or negative/positive SAT. You can see the changes in pages 5-7, lines 210-219.
Comments 6: [In the discussion, authors should make it clear when they are discussing the results of the present study. For example, in lines 221-224, are the results from other studies?]
Response 6: Thank you for pointing this out. We agree with this comment. Careful corrections were made regarding this in the discussion section. For example, in the example you gave ‘In this study’ removed and amended with ’And the researchers of that study found that’ in page 8, line 265
Comments 7: [Check for minor typographical errors: datas (line 70), gr (line 93).]
Response 7: Thank you for pointing this out. We agree with this comment. The errors you mentioned were corrected. In page 2, line 88; and in page 3, line 132.
Comments 8: [Check that Brucella should be in italics when applicable.]
Response 8: Thank you for pointing this out. We agree with this comment. All words relating to the genus name ‘Brucella’ were changed to italics in the manuscript.
Reviewer 3 Report
Comments and Suggestions for Authors
General Comments
-Introduction / Background:
- The authors should provide a more complete overview of brucellosis, including classic symptoms, typical diagnostic methods, and standard treatments.
- It is unclear whether the treatments used in this study are routine or novel, or whether the clinical findings observed are typical of brucellosis or represent unusual features.
-Formatting / Style:
- The genus name Brucella should be italicized throughout the manuscript according to standard nomenclature.
Specific Comments
-Line 21: AST and ALT are mentioned without explanation. A brief description of what they are (liver enzymes) and their relevance in brucellosis would improve clarity.
-Lines 155–160: Paragraphs are long, repetitive, and somewhat confusing. For example, the description of AST and ALT results could be simplified: "Among patients without a decrease in SAT values, mean post-treatment AST and ALT levels were significantly higher compared with those in the decreased SAT group (p<0.001 and p=0.049, respectively). Although not statistically significant, a similar tendency was observed before treatment, with patients in the non-decreased SAT group showing higher baseline AST and ALT levels than those with decreased SAT values (Table 2)." This formulation reduces redundancy while maintaining the key message.
-Lines 179–188: The first paragraph of the discussion is very repetitive, essentially restating results already presented in the Results section. It could be summarized more concisely: "In this study, Brucella SAT values remained positive in a substantial proportion of patients after treatment. Higher initial SAT values (≥1/320) were associated with persistent SAT positivity, while patients with lower initial titers (≤1/160) were more likely to achieve serological negativity. AST and ALT levels tended to remain elevated in patients whose SAT values did not decrease."
Author Response
Response to Reviewer 3 Comments
Thank you very much for taking the time to review this manuscript. Please find the detailed responses below and the corresponding revisions/corrections highlighted yellow in the re-submitted files. The corrections proposed by other reviewers are highlighted by different colours. Thank you again for your suggestions on how to improve the readability and value of the article, and for giving us the opportunity to implement them. All your suggestions were reviewed point-by-point, and the manuscript was revised taking them into consideration. General information regarding brucellosis was incorporated into the introduction section, and the sentences you suggested were utilised to enhance the presentation of the results. Additionally, the text was improved in general and spelling errors were corrected. If you have any suggestions when you review the manuscript, please do not hesitate to let us know, we would be grateful if you could share them with us.
Point-by-point response to Comments and Suggestions for Authors
Comments 1: [The authors should provide a more complete overview of brucellosis, including classic symptoms, typical diagnostic methods, and standard treatments.]
Response 1: Thank you for pointing this out. We agree with this comment. We have accordingly revised to emphasize this point. Transmission routes, clinical symptoms, diagnosis, laboratory characteristics and standard treatments were mentioned in the introduction section of the manuscript. Page 1, lines 35-48 as ‘This zoonosis is endemic in the Mediterranean region, Latin America, Africa, the Middle East, and Western Asia. The causative agent of the disease is Brucella spp.; Gram-negative, oxidase-positive, urease-positive coccobacillus. The bacteria are present in significant quantities in the milk, urine, and birth secretions of infected animals; causing mild or asymptomatic infections. It can be transmitted through direct contact with infected animals' tissues or secretions, consumption of unpasteurized dairy products, and inhalation of aerosols1,2. The symptoms of brucellosis are non-specific. The most common symptoms include fever, chills, fatigue, loss of appetite, weight loss, headache, night sweats, myalgia and arthralgia. In the event of localised involvement, symptoms specific to the affected body part are observed 2,3. In cases of brucellosis, there is a possibility of mild transaminase elevation and the subsequent development of hepatitis. Additionally, a range of hematological abnormalities, including anaemia, lymphocytosis, leukocytosis, leukopenia, and thrombocytopenia, have been documented4,5.’ and page 2, lines 50-60 as ‘Blood, tissue or bone marrow culture are gold standard methods for the diagnosis of brucellosis. In whom the culture resulted negative but there is a clinical suspicion, and in areas where the culture is not possible, serological tests are crucial for confirming the diagnosis of brucellosis. Agglutination tests are a type of serological test used to assess antibody responses. These tests examine the direct agglutination of bacterial antigens with specific antibodies in the patient's serum2,7. According to the World Health Organization (WHO)'s laboratory criteria for the diagnosis of brucellosis, the isolation of Brucella spp. or a standard tube agglutination test (SAT) titer of 1/160 and above is required8. Brucellosis is an intracellular pathogen, so it requires long-term combination therapy with multiple antibiotics to prevent recurrence and chronicity.’ Detailed treatment regimens were also mentioned in method section.
Comments 2: [It is unclear whether the treatments used in this study are routine or novel, or whether the clinical findings observed are typical of brucellosis or represent unusual features.]
Response 2: Thank you for pointing this out. We agree with this comments. The treatment applied to patients were suggested therapies in the global by the experts. This was explained in the method section under the subheading ‘Treatment regimens and duration of therapy’. In page 3, line 126-131 as ‘The patients received standard therapy with at least two drugs (doxycycline + streptomycin or doxycycline + rifampicin), as recommended by some experts and WHO3,4. Clinician-based treatment was administered to patients with osteoarticular involvement and to some patients predicted to be unresponsive to therapy by adding a third agent to doxycycline + rifampicin. The doses and the administration route of the drugs used in the treatment of brucellosis:’. Whether the clinical findings were typical or not was mentioned in method section under the subheading ‘Definitions’ in page 3, lines 101-15 as ‘Patients with SAT titers ≥1/160 and a range of symptoms including fever, chills, fatigue, loss of appetite, weight loss, headache, night sweats, myalgia and arthralgia lasting less than a year were considered to have acute brucellosis. Despite the fact that the majority of these symptoms were not specific to brucellosis, they were considered as brucellosis symptoms because they could not be explained by any other causes.’
Comments 3: [The genus name Brucella should be italicized throughout the manuscript according to standard nomenclature.]
Response 3: Thank you for pointing this out. We agree with this comment. All words relating to the genus name ‘Brucella’ were changed to italics in the manuscript.
Comments 4: [Line 21: AST and ALT are mentioned without explanation. A brief description of what they are (liver enzymes) and their relevance in brucellosis would improve clarity.]
Response 4: Thank you for pointing this out. We agree with this comment. AST and ALT were explained in abstract section, page 1, lines 20-21 as ‘liver enzymes such as aspartate aminotransferase (AST) and alanine aminotransferase (ALT)’.
Comments 5: [Lines 155–160: Paragraphs are long, repetitive, and somewhat confusing. For example, the description of AST and ALT results could be simplified: "Among patients without a decrease in SAT values, mean post-treatment AST and ALT levels were significantly higher compared with those in the decreased SAT group (p<0.001 and p=0.049, respectively). Although not statistically significant, a similar tendency was observed before treatment, with patients in the non-decreased SAT group showing higher baseline AST and ALT levels than those with decreased SAT values (Table 2)." This formulation reduces redundancy while maintaining the key message.]
Response 5: Thank you for pointing this out. We agree with this comment. We have accordingly revised to emphasize this point. The expression you suggested has been added to results section in page 5, lines 196-201 as ‘Among patients without a decrease in SAT titers, mean post-treatment AST and ALT levels were significantly higher compared with those in the decreased SAT group (p<0.001 and p=0.049, respectively). Although not statistically significant, a similar tendency was observed before treatment, with patients in the non-decreased SAT group showing higher baseline AST and ALT levels than those with decreased SAT titers (Table 1).’
Comments 6: [Lines 179–188: The first paragraph of the discussion is very repetitive, essentially restating results already presented in the Results section. It could be summarized more concisely: "In this study, Brucella SAT values remained positive in a substantial proportion of patients after treatment. Higher initial SAT values (≥1/320) were associated with persistent SAT positivity, while patients with lower initial titers (≤1/160) were more likely to achieve serological negativity. AST and ALT levels tended to remain elevated in patients whose SAT values did not decrease."]
Response 6: Thank you for pointing this out. We agree with this comment. We have accordingly revised to emphasize this point. The expression you suggested has been added to discussion section as a first paragraph in page 7, lines 221-225 as ‘In this study, Brucella SAT titers remained positive in a substantial proportion of patients after treatment. Higher pre-treatment SAT titers (≥1/320) were associated with persistent SAT positivity, while patients with lower pre-treatment SAT titers (≤1/160) were more likely to achieve serological negativity. AST and ALT levels tended to remain elevated in patients whose SAT titers did not decrease.’
Round 2
Reviewer 1 Report
Comments and Suggestions for Authors
All of our observations were accepted or adequately answered. Therefore, we suggest that the manuscript is ready for publication.